# Protocol of the Long-COVID Patients Causal Diagnosis and Rehabilitation Randomized Feasibility Controlled Trial in Patients with Dysautonomia: The LoCoDiRe-Dys Study

Dimitrios Spaggoulakis [1,*], Antonios Kontaxakis [2,*], Andreas Asimakos [1], Stavroula Spetsioti [1], Archontoula Antonoglou [1], Pantelis Gounopoulos [3], Martha Katsarou [4,5], Helen Iasonidou [3], Stergios-Stylianos Gatzonis [6] and Paraskevi Katsaounou [1]

[1]   First Department of Critical Care Medicine and Pulmonary Services, Evangelismos Hospital, National and Kapodistrian University of Athens, 10676 Athens, Greece

[2]   Physical Medicine & Rehabilitation Department, 414 Military Hospital of Special Diseases, 15236 Penteli, Greece

[3]   2nd Cardiology Department, "Evangelismos" Hospital, 10676 Athens, Greece

[4]   LONG COVID GREECE (Official Society), 54636 Thessaloniki, Greece

[5]   Research Group of Clinical Pharmacology and Pharmacogenomics, Faculty of Pharmacy, School of Health Sciences, National and Kapodistrian University of Athens, 15771 Zografou, Greece

[6]   1st Department of Neurosurgery, "Evangelismos" Hospital, National and Kapodistrian University of Athens, 10676 Athens, Greece

*   Correspondence: dimitrisspglks@gmail.com (D.S.); akontaxakis@yahoo.gr (A.K.)

**Abstract:** Dysautonomia in the post-COVID-19 condition appears to affect a significant number of patients, with reports raising the incidence up to 61%, having an overlap with myalgic encephalomyelitis/chronic fatigue syndrome. Quality of life and daily function are significantly impacted and conservative management interventions, despite the lack of high-quality evidence to date, are needed to ameliorate disability. A total of 50 adults with a dysautonomia post-COVID-19 diagnosis based on the Ewing battery and a NASA lean test will be enrolled in a randomized single blinded controlled trial with a crossover design. Feasibility and lack of definite dysautonomia diagnosis will be the primary outcomes, while secondary outcomes will be health-related, clinical and cardiopulmonary exercise test indicators. Safety and acceptance will also be checked, primarily excluding participants with post-exertional malaise. The Long-COVID patients Causal Diagnosis and Rehabilitation study in patients with Dysautonomia (LoCoDiRE-Dys) intervention will consist of an educational module, breathing retraining and an individualized exercise intervention of biweekly sessions for two months with regular assessment of both groups. LoCoDiRe-Dys aims to be the first post-COVID-19 randomized study in people with dysautonomia offering a multimodal intervention both in diagnosis and management. The need for evidence in effectively supporting patients is eminent.

**Keywords:** dysautonomia; feasibility; post-COVID-19 condition; randomized controlled trial; rehabilitation; SARS-CoV-2

## 1. Introduction

While most measures concerning the SARS-CoV-2 pandemic are being lifted, humanity counts a toll of approximately 7 million deaths and 760 million people infected as being reported by the World Health Organization (WHO) ([1] https://COVID19.who.int/, accessed on 15 April 2023). The acute phase of the pandemic may be over, still data are building up on the persistent complications of the post-COVID-19 condition regardless of hospitalization, sex, age and comorbid conditions [2] with a wide range of persistence being reported (15.1–58.9%) [2–5] one year post-infection. The estimates for the number of people affected rise up to 17 million in Europe as addressed by the WHO [6], with over

6% of the adult population in the US being currently reported by the National Center for Health Statistics [7].

Over 200 symptoms have been recognized [8], and the involvement of the autonomic nervous system is noteworthy. Dysautonomia is found in 9–61% of people with post-COVID-19 condition [9] and despite the diverse methodology used in research, the burden of function of such disorders is significant [10]. Although the dysautonomia term has justifiably been criticized as a non-specific one [11], specific diagnoses include Orthostatic Intolerance (OI) (with included sub-diagnoses of postural orthostatic tachycardia, neurogenic orthostatic hypotension, vasovagal syncope) and inappropriate sinus tachycardia (IST) [12].

While the scope of this article is not delving into the clinical presentation of all of them which may be overimposed by each other [13], orthostatic hypotension appears with a fall in systolic pressure >20 mmHg or >10 mmHg in diastolic blood pressure upon resumption of standing [14], while in inappropriate sinus tachycardia, resting heart rate is >100 without regard to positional change [15]. As far as the most extensively studied condition, postural orthostatic tachycardia is typically defined as an increase in heart rate of $\geq$30 upon standing in adults accompanied by objective symptoms without orthostatic hypotension [16]. Apart from palpitations, chest pain, nausea, cognitive impairment, generalized weakness and visual disturbances have been described [17]. Adding to the complexity, dysfunctional breathing has also been reported to a significant proportion of POTS patients [18], indicating the need of a multimodal assessment and intervention. The etiology is multifaceted including peripheral autonomic denervation, hypovolemia, autoimmune, mastocyte activation syndrome [13,16], which are also involved in the mechanisms implicated in a post-COVID-19 condition [19].

As far as management is concerned, pharmacological measures with rhythm reduction through b-blockers and ivabradine, vasoconstriction through midodrine, volume expansion through fludrocortisone and others while non-pharmacological interventions have been well described [16,20]. Despite the lack of high-quality evidence especially for the non-pharmacologic ones including reclined aerobic and strengthening training, they are suggested as first line interventions [16]. On the other hand, data on post-COVID-19 condition dysautonomia interventions are drawn from former conditions [12], while evidence remains scarce.

Still autonomic manifestations are not isolated from other conditions as they have also been implicated in another post-infectious one, myalgic encephalomyelitis/chronic fatigue syndrome, where over 90% appear with orthostatic intolerance symptoms [21] apart from the main feature which is fatigue and delayed symptom exacerbation (post-exertional malaise—PEM). This condition is all the more being recognized in a significant proportion of post-COVID-19 condition patients [19] with Jason and Dorri reporting a prevalence of 58% [22].

Regarding exercise interventions in this population, of note is that instead of an increase in the anaerobic threshold through aerobic training, quite the opposite results in a 2-day cardiopulmonary exercise test (CPET) with a significant relapse of fatigue of over 2 weeks [23]. At the same time, evidence for reduced systemic oxygen extraction in invasive CPET [24] and increased lactate levels with minimal exertion in post-COVID-19 condition patients [25] further underline similar findings with me/cfs [26] and question the role of exercise in dysregulated muscle energy consumption. Therefore, as it has clearly been delineated in expert opinion consensus [12], the WHO clinical guidance [27] as well as patient lead organizations [28], exercise-centered interventions need to be carefully designed and pursued where appropriate.

Why is such a trial needed?

Taking in consideration the complexity of the multisystemic effects of SARS-CoV-2 [19], and the non-homogenous population of people with a post-COVID-19 condition, it is critical to adapt rehabilitation interventions offered, especially when taking in consideration the vast amount of patients that are expected to already live in the community. The

recommendations offered above are based on mostly observational studies [29–33] mainly focusing on exercise training. What is missing to date, apart from the design of the trial, is a multicomponent intervention, including a greater array of respiratory interventions and a broader dysautonomia population. Adding to the literature, we are intending to provide evidence of the feasibility of a personalized rehabilitation intervention of non-ME/CFS dysautonomia in a post-COVID-19 condition.

## 2. Materials and Methods

### 2.1. Study Design

This study is a feasibility randomized unblinded controlled trial with 2 arms. Patients participating will be visiting the outpatient post-COVID-19 condition clinic of the Pulmonary Dept- First ICU National and Kapodistrian University of Athens experiencing dysautonomia. Among those patients standard diagnostic procedures will be carried out and personalized treatment including adequate medications will be given. Recruitment has begun from February 2023. This study is registered on ClinicalTrials.gov (NCT05855356) based on the SPIRIT (Standard Protocol Items: Recommendations for Interventional Trials) guidelines (see Supplementary File).

### 2.2. Eligibility Criteria

Adult patients with proven or suspected SARS-CoV-2 infection as per the WHO post-COVID-19 condition (associated symptoms beyond 3 months from infection onset persisting for 2 months) and dysautonomia will be included. See Table 1 for more details.

**Table 1.** Study inclusion and exclusion criteria.

| Inclusion Criteria | Exclusion Criteria |
| --- | --- |
| Adults 18–65 years of age | Absolute or relative contraindications to exercise due to cardiac pathology |
| * WHO definition of post-COVID-19 condition | Serious mental/cognitive impairment that will not allow systematic participation |
| Confirmed dysautonomia diagnosis through Ewing Battery | Unable to regularly reach the center |
| Able to attend 2 times/week for 8 weeks | Pregnancy |
| | * CFS/ME fulfilling the Canadian Consensus Criteria |
| Able to provide informed consent | Secondary health conditions that would explain symptoms, intervene in dysautonomia diagnosis or would impede participation in the exercise protocol (i.e., untreated hypothyroidism and Diabetes Melitus, major psychiatric disorders, * COPD, * PICS, pulmonary fibrosis, chronic respiratory or heart failure, not ambulatory, suffering from dementia, chronically paralyzed, with paraplegia, with multiple injuries or other serious orthopedic problems that caused disability, patients suffering from very serious underlying diseases such as end-stage cancer, and those with neurological diseases causing disability) |

Abbreviations: * WHO = World Health Organization, * CFS/ME = chronic fatigue syndrome/myalgic encephalomyelitis, * COPD = chronic obstructive pulmonary disease, and * PICS = post-intensive care syndrome.

### 2.3. Ethics

This study has been approved by the Ethics Committee of the Evangelismos hospital and is being conducted in accordance with the Declaration of Helsinki [34]. Written informed consent was required and participation could be stopped at any time.

### 2.4. Study Population

Patients attending the outpatient clinic which is in close collaboration with the Greek Long COVID Patient Organization with referrals throughout Greece. Further, patients who were hospitalized in the General Hospital of Evangelismos can be included.

*2.5. Study Procedures*

2.5.1. Clinical Assessment

A thorough clinical assessment will take place, which has already been published elsewhere [35] so as to exclude co-existent pathology. Dysautonomia will be characterized based on definite autonomic nervous system involvement through the Ewing battery [36], which entails two or more of the heart rate tests being abnormal, with tests (iv) and (v) as follow being used for the definition of severe involvement. Cardiovascular tests [36] will be measured both at the beginning and end of the intervention as follows:

(i)     The Valsalva maneuver, where the patient blows in a mouthpiece at a pressure of 40 mmHg for 15 s and an ECG measurement takes places of the ratio of the longest R-R interval after the maneuver with the shortest during the maneuver ("Valsalva ratio" being the mean of 3 subsequent maneuvers).

(ii)    The heart rate response to standing up, where the patient lies and then stands up without help and a ratio of the R-R interval is measured (30th beat to 15th beat after standing up).

(iii)   The heart rate response to deep breathing, where the patients slows his breathing rate to 6 breaths/min and the maximum and minimum heart rates during each breath are measured (mean difference of three successive attempts is used).

(iv)    * The blood pressure response to standing up, with measures taken between lying and standing position (difference of systolic blood pressure is measured).

(v)     * The blood pressure response to handgrip, with the patient maintain 30% of the maximum voluntary contraction with a handgrip at max 5 min (difference of diastolic pressure close to releasing and before the beginning of the test).

Accompanying, NASA lean test [37] (comprising a ten-minute observation of blood pressure and heart rate) will be performed to distinguish between neurogenic orthostatic hypotension, postural orthostatic tachycardia syndrome and inappropriate sinus tachycardia syndrome.

Further, in order to exclude participants that are prone to exacerbation after exercise interventions, existence of ME/CFS-like condition will be checked through the Canadian Consensus Criteria [38] while subsequent post-exertional malaise will be assessed through the Post-Exertional Malaise subscale of the DePaul Symptom Questionnaire [39].

2.5.2. Functional Assessment

During the functional assessment of the participants, a standardized 6 minute walk test [40] will take place, with continuous measurement of oxygen saturation, heart rate, respiratory rate and degree of exertion through the modified Borg scale (0–10). Further, the 1 minute sit to stand test [41] and muscular strength test of upper and lower limbs will take place. As far as balance is concerned, it will be assessed through the Activities-Specific Balance Confidence (ABC) Scale [42]. In parallel, assessment of dysfunctional breathing pattern will also be performed through manual assessment of respiratory motion [43] by a physiotherapist.

2.5.3. Outcome Measures

The following outcome measures will be used before the intervention and at specified times throughout this study.

(i)     Breathing function and dyspnea through the modified Medical Research Council scale [44] and the Nijmegen Questionnaire [45,46]

(ii)    Fatigue severity through the fatigue severity scale [47,48]

(iii)   Physical activity level with the International Physical Activity Questionnaire [49,50]

(iv)    Cognitive function through the Montreal Cognitive Assessment [51,52]

(v)     Assessment of quality of life through the European Quality of Life—5 dimensions—5 Levels questionnaire [53,54]

(vi) Assessment of emotional function through the Hospital Anxiety and Depression Scale [55,56]

### 2.5.4. Cardiopulmonary Exercise Test

Following a 3-min rest period, each subject will undergo a ramp-incremental test on an electromagnetically braked cycle ergometer (Viasprint 150 P; Vyaire Medical GmbH, Hoechberg) until they reach their limit of tolerance. The test includes 3 min of unloaded pedaling, followed by 1-min increments of 5–20 watts. The highest work level reached and maintained at a pedaling frequency of 60 revolutions per minute for 30 s is recorded as peak work rate (PWR).

During the test, subjects wear a nose clip and breathe through a mouthpiece while pulmonary gas exchange and ventilatory variables are measured using calibrated signals from gas analyzers and a mass flow sensor (Vmax 229; Vyaire Medical GmbH). Breath-by-breath measurements included pulmonary oxygen uptake (VO2), pulmonary carbon dioxide output (VCO2), respiratory exchange ratio, minute ventilation (VE), tidal volume (VT), and respiratory frequency (fR). Cardiac frequency (fC) is determined using the R-R interval from a 12-lead on-line electrocardiogram and SaO2 by pulse oximetry (SpO2). Shortness of breath was rated every 2 min using the 0–10 Borg category ratio scale [57].

Peak VO2 from the ramp-incremental test is compared with that from the JONES method [58]. The slope of VO2/WR was also calculated, and the V-slope technique was used to detect the VO2 at which the lactate threshold (LT) occurred [59]. The LT is identified independently by two observers on both pre- and post-training exercise data sets.

During exercise testing, dynamic inspiratory capacity (IC) is measured every 3 min and at the end of exercise to evaluate changes in operational lung volumes. Inspiratory reserve volume (IRV) is calculated by subtracting VT from the coinciding IC. Prior to exercise testing, patients are familiarized with the IC maneuver and instructed to make 3–5 maximal efforts. Changes in end-expiratory and end-inspiratory lung volumes (EILV) are assumed to reflect changes in IC, assuming that total lung capacity (TLC) remained constant during exercise [60].

Following the CPET, all patients are given again the Post-Exertional Malaise subscale of the DePaul Questionnaire and, if positive for PEM, are excluded from this study and further rehabilitation advice is given as regards to ME/CFS management.

### 2.5.5. Primary and Secondary Outcomes

Primary outcomes will be feasibility and lack of definite dysautonomia diagnosis. All of the rest objective and patient reported measures including health-related, clinical and cardiopulmonary exercise test indicators will be reported as secondary outcomes. All of these data will be anonymized and secured based on GDPR regulations in the department's computer.

### 2.5.6. Sample Size Calculation

Being a feasibility trial, recruitment and adherence will mainly be assessed, so sample size is not intended on power calculation. Still, based on the only RCT to date with a main focus on OI in which 62.5% of the intervention participants no longer satisfied the OI criteria with 9% of the control group [29] with difference being significant (Fisher's exact test: $p = 0.008$, 2-sided) and for an assumed 10% spontaneous remission rate and a healing rate approximately 60% of the test power would be 80%. Taking in consideration a large drop out rate in our study due to the overlap with ME/CFS of 40%, the total sample size is calculated at 50. Taking into account the vast waiting list for the outpatient clinic for over 3 months, this number of patients is regarded plausible. Termination of this study will be decided by the study supervisor when the number will be met or in case of unexpected adverse events.

### 2.5.7. Randomization and Design

Simple block randomization was used as described by Parker et al. [61] through the algorithm provided by http://www.randomization.com (accessed on 15 April 2023) into blocks of 4 and 6. A separate study coordinator was responsible for attributing each participant enrolled in either the rehabilitation intervention or the standard of care one. Participants will be allocated to the LoCoDiRe-Dys intervention and the standard of care arm by the exercise physiologist or the physiotherapist and evaluated at the beginning of this study, in 8 weeks at the end of the intervention and at 16 weeks follow-up as in Figure 1. To ensure that all participants will receive rehabilitation treatment, the control group participants will be offered to participate in the rehabilitation regimen if interested at the end of this study.

| | Study Period | | | | |
|---|---|---|---|---|---|
| | Enrolment | Allocation | Post Allocation | | |
| TIMEPOINT** | $-t_1$ | 0 | $t_1$ <br> *Week 0* | $t_2$ <br> *Week 8* | $t_2$ <br> *Week 16* |
| **ENROLMENT:** | | | | | |
| **Eligibility screen** | X | | | | |
| **Informed consent** | X | | | | |
| **Randomization** | 50 | | | | |
| **Allocation** | | X | | | |
| **INTERVENTIONS:** | | | | | |
| *LoCoDiRe-Dys* | | X | | | |
| *Standard of Care* | | X | | | |
| **ASSESSMENTS:** | | | | | |
| *Ewing Battery* | | | X | X | X |
| *NASA Lean Test* | | | X | X | X |
| *6MWT/1mSTS** | | | X | X | X |
| *FSS** | | | X | X | X |
| *mMRC Dyspnea* Scale* | | | X | X | X |
| *Nijmegen Questionnaire* | | | X | X | X |
| *MoCA** | | | X | X | X |
| *IPAQ** | | | X | X | X |
| *HADS** | | | X | X | X |
| *EQ-5D 5L** | | | X | X | X |
| *CPET** | | | X | X | X |
| *Sociodemo- graphic data* | | | X | | |

**Figure 1.** Schedule depicting enrollment and interventions of the LoCoDiRe-Dys study based on SPIRIT 2013 statement [62], * 6MWT = 6 minute walk test, FSS = fatigue severity scale, mMRC Dyspnea Scale = modified Medical Research Council Dyspnea scale, MoCA = Montreal Cognitive Assessment, IPAQ = International Physical Activity Questionnaire, HADS = Hospital Anxiety and Depression Scale, EQ-5D 5L = EuroQoL 5 Dimensions 5 Levels, and CPET = Cardiorespiratory Exercise Test, ** $-t_1$ = pre-allocation, $t_1$ = beginning of the study, $t_2$ = end of intervention, $t_3$ = 2 month follow up.

### 2.5.8. Study Intervention

The rehabilitation intervention consists of breathing retraining, aerobic and strength training supervised by the department's exercise physiologist and physiotherapist; and personalized according to the F.I.I.T. (frequency, intensity, time, type) principle following the current recommendations for the post-COVID-19 condition [27]. The introductory session of breathing retraining (see Supplementary File) with advice for 2–10 min practice at home, will be followed by biweekly aerobic interval exercise training on a semi-recumbent cyclo-ergometer as tolerated with the aim of 30 min duration at 60% of HRmax and correspondingly progress to upright aerobic exercise. During the cool down phase breathing retraining will also shortly be reminded and take place. Afterwards in the same sessions, strength training will follow, with a focus on lower body and core with a starting 10 min duration with 2 sets of 10 repetition of seated leg press, leg curl, leg extension, calf raise, chest press and seated row with increasing wait and duration till 30 min as tolerable. Both aerobic and strengthening training were adapted from former studies (Fu and Levine [63]) so as to watch for the emergence of PEM which will be assessed with the DePaul Questionnaire subscale for PEM before each session. All patients will be educated regarding the importance of incorporating conservative measures that would be of help especially for postural orthostatic tachycardia syndrome (Table 2).

**Table 2.** Measures offered in conservative treatment of people with dysautonomia.

| | Conservative Measures |
|---|---|
| 1 | 3–4 liquid intake per day |
| 2 | 500 mL of water before getting out of bed in the morning and before exercise |
| 3 | 8–12 g of salt daily |
| 4 | Graduated compression garment (ideally class II (20–30 mmHg) covering the lower body and/or an abdominal binder based on tolerance) |
| 5 | Leveling up the head of the bed by 10–18 cm with a bed riser or a full length wedge underneath the mattress. |
| 6 | Maneuvers that increase venous return in case of discomfort/dizziness when standing (leg crossing, isometric contraction of lower limbs, abrupt coughing, squatting) |
| 7 | Avoidance of large meals rich in fat-favoring smaller regular meals |
| 8 | Avoidance of high temperatures, caffeine, and alcohol intake |
| 9 | Exercise in the seated position or swimming |
| 10 | Medication as indicated |

### 2.5.9. Adverse Events and Drop-Out Monitoring

Adverse events (AEs) will be recorded as well as reasons for drop-out, while any patient can withdraw at any time. Any medical event that will happen during this study will be regarded as an AE, such as fatigue and especially post-exertional malaise, soreness, pre-syncope, muscle and/or joint pain or unexpected regarding any other complication. The Scientific Committee overseeing the project will be informed for any unexpected event accompanied by data on the relation to the intervention.

### 2.5.10. Patient Organization Feedback

The whole study is involving members of administrative committee of the Long COVID Greece® patient organization, at every step of the design and implementation. Feedback is invaluable so as to carefully offer an intervention that would meet the lived experience needs and provide safe and effective guidance to participants.

### 2.5.11. Data Collection and Analysis

Analyses will be performed using SPSS (Ver 25. IBM Corp., Chicago, USA). Prior to analysis, the assumption of normality for outcomes will be assessed using the Shapiro Wilk Test. Descriptive statistics included mean ($\pm$SD or SE) or median (Interquartile Range-IQR) as appropriate. Comparisons of baseline characteristics between the intervention

and UC groups will be made using independent sample *t*-test or Mann–Whitney U tests for non-parametric data. Within group differences will be assessed by paired t-tests for parametric data or Wilcoxon signed rank tests for non-parametric data. Fisher's exact test will be used for categorical data. Univariate analysis of variance (ANCOVA) will be used to compare Rehab and Standard of Care groups at two and four months using baseline values (including patients that will be lost at follow up) as covariate to compensate for potential baseline differences and missing data. Multiple comparisons will be adjusted using the Bonferroni method. Non-parametric ANCOVA (Quade's) will be used for non-parametric data [64]. Statistical significance will be set at $p < 0.05$ for all analyses.

## 3. Discussion

The LoCoDiRe-Dys study has been developed with the aim of providing data in the feasibility of safe rehabilitation interventions to people with post-COVID-19 dysautonomia building upon interventions not systematically offered due to the lack of well-designed trials till now. Some prospective and retrospective cohorts have already showed effectiveness specifically in the postural tachycardia syndrome [30–33] with just one RCT [29] to our knowledge, with no specific adverse events described. None till now address the post-COVID-19 condition population and none attempts have been made to cover a greater population meeting the dysautonomia criteria.

It is to be noted that such interventions are rarely offered, especially in facilities and health care systems where such dedicated clinics are sparse if not absent. Collaboration with the Long COVID Greece patient organization is also a means to incorporate patient feedback and lived experience in each phase of this study, at the same time providing smooth knowledge transmission in the regional healthcare system as well as a platform to communicate results with patients.

Rehabilitation interventions have widely been introduced early in the beginning of the pandemic both in acute and post-acute/chronic stage of COVID-19 with a large amount of evidence being gathered regarding its effectiveness [65]. At the same time, special interest in telerehabilitation is also rising with equal preliminary data of effectiveness [66]. Experience gathered till now is deemed to be modified and translated in the dysautonomia population, with the hope that services will be correspondingly developed.

Study limitations are the lack of a specialized autonomic laboratory that would have made more detailed diagnosis, the inherent difficulty in implementing a double blinded intervention in rehabilitation, the length and number of sessions that were chosen due to staffing limitations. Further, the difficulty in stratification of patients based on medication offered which will be controlled during the feasibility stage. Suggestions for future research will be better laboratory and dysautonomia specific investigation leading to the root causes of dysautonomia, larger multicenter rehab interventions adjusted for etiology and medical interventions, compliance with conservative measures and better differentiation from the CFS/ME subpopulation.

The LoCoDiRe-Dys study will provide data on the feasibility of a personalized rehabilitation intervention so that a greater multicenter intervention will be made possible within different post-COVID-19 clinics.

**Supplementary Materials:** The following supporting information can be downloaded at: https://www.mdpi.com/article/10.3390/biomed3030026/s1, SPIRIT [62] checklist.

**Author Contributions:** Conceptualization, D.S. and A.K.; methodology, A.A. (Andreas Asimakos); validation, S.S.; investigation, A.A. (Archontoula Antonoglou), D.S., A.A. (Andreas Asimakos), S.S. and P.G.; resources, D.S.; data curation, S.S.; writing—original draft preparation, A.K.; writing—review and editing, D.S., A.A. (Andreas Asimakos), M.K., H.I. and S.-S.G.; visualization, A.K.; supervision, P.K.; project administration, P.K.; funding acquisition, not applicable. All authors have read and agreed to the published version of the manuscript.

**Funding:** This research received no external funding.

**Institutional Review Board Statement:** This study was conducted in accordance with the Declaration of Helsinki, and approved by the Institutional Review Board (or Ethics Committee) of General Hospital of Athens "Evangelismos" (protocol code 22 and date of approval 22 January 2021). All results will be published according to CONSORT [67] guidelines.

**Informed Consent Statement:** Informed consent will be obtained from all subjects involved in this study by the exercise physiologist and the physiotherapist.

**Data Availability Statement:** Data and material will be available upon completion of this study on request.

**Conflicts of Interest:** The authors declare no conflict of interest.

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
