# Peer review of "Protocol of the Long-COVID Patients Causal Diagnosis and Rehabilitation Randomized Feasibility Controlled Trial in Patients with Dysautonomia: The LoCoDiRe-Dys Study"

_2673-8430, doi:10.3390/biomed3030026_

Round 1
Reviewer 1 Report
This article reported the protocol of the Long-CoViD patients causal diagnosis and rehabilitation randomized feasibility controlled trial in patients with Dysautonomia, which is an interesting topic today and well done. However, some concerns in this article need to be addressed.
1. Please write keywords in a consistent format (first letter).
2. In the first citation, put the full name before the word abbreviation, followed by the abbreviation (e.g. covid-19). Please correct.
3. Use lowercase letters at the beginning of words in sentences.
4. Please enter your questionnaire as a supplementary file.
5. On line 40 add the WHO link for your reference.
6. Please use the WHO link instead of reference 5.
7. Please change Long-CoViD to Long COVID.
8. For better understanding, please explain the abbreviations in Table 1 below.
9. Please discuss further and improve in the discussion section.
10. Research limitations and suggestions were not addressed at the end of the discussion. Please include these.
11. Update the references.
Minor editing of English language required.
Author Response
Thanks so much for the feedback.
1)Changed to alphabetical order
2) Didn't quite comprehend- is there a need to refer to coronavirus disease 2019 (covid-19)? Didn't change anything there as covid-19 is widely used.
3) Do you mean that we should change the first letter of each sentence from capital to lower case?
4) Demographics questionnaire included as supplementary
5) Added as a reference too
6) Estimate was not found as an official WHO site/ statement but as referenced by Dr Wise in his BMJ publication, as referenced
7) Long COVID corrected
8) Abbreviations included
9) Discussion was extended
10) Done
11) Done
Reviewer 2 Report
Dear Author,
thank you for sharing this study protocol. After a careful revision, I found it well written. No changes are needed.
We waiting for the results.
Regards.
Author Response
Thanks so much for the review!
Reviewer 3 Report
An educational and informative manuscript that has both clinical interest and merit. However, there are some editing issues that the authors should consider and address. The following are suggestions/comments regarding those issues. Line 46, "adult population in the US being currently ...". Line 48, "... have been recognized [7], the involvement of the ....". Lines 51 & 52, "... the dysautonomia term has justifiably been ...". Line 59, "... of standing [13], while in ...". Line 66, "patients [17], indicating the need ...". Line 69, "... in a post covid-19 ...". Line 75, "On the other hand, data on post ...". Line 82, "This condition is all the more being ...". Line 90, "patients [24] further displays such findings with ...". Line 97, "... population of people with a post covid-19 ...". Line 105, "... dysautonomia in a post covid-19 condition." Line 136, "... will take place which has already been published ...". Line 172, "In parallel, assessment of dysfunctional ....". Line 191, "... each subject will undergo a ramp-incremental test ...". Line 218, "... given as regards to ME/CFS management." Line 223, "and Cardiopulmonary Exercise Test indicators ...". Lines 230 & 231, "... no longer satisfied the OI ...". Line 233, "... rate about 60% of the test power ...". Lines 235 & 236, "... outpatient clinic for over 3 months, this number ...". Line 242, "... was used as described by Parker et al. [60]...". Lines 243 & 244, "...into blocks of 4 & 6. A separate study ...". Line 247, "physiologist or the physiotherapist and ...". Line 261, "... and physiotherapist; and personalized ...". Line 263, "... recommendations for any post covid-19 ...". Line 264, "... with advice for 2- 10 min practice ...". Line 271, "seated row with increasing wait and ...". Line 272, "... former studies (Fu & Levine, [62]), so as to ...". Line 283, " ... can withdraw at any time." Line 284, "... regarded as an AE, such as fatigue and ...". Lines 292 & 293, "... that would meet the experience needs and ...". Lines 312 & 313, "... dysautonomia and to build upon interventions ....". Line 322, "feedback and live experiences in each phase of ...". Line 326, "... intervention so that a greater multicenter ...".
Well written; however, the manuscript has some editing issues.
Author Response
Thank you so much for your kind review!
All comments have been included apart from the last ones which deal with the lived experience of patients, underlining both their experience throughout the study as well as effects in their everyday life.